# Role of Sigma-1 Receptor in Calcium Modulation: Possible Involvement in Cancer

**DOI:** 10.3390/genes12020139

**Published:** 2021-01-22

**Authors:** Ilaria Pontisso, Laurent Combettes

**Affiliations:** 1UMR 1282, INSERM, Laboratoire de Biologie et Pharmacologie Appliquée, Ecole Normale Supérieure Paris Saclay, 91190 Gif Sur Yvette, France; ilaria.pontisso@universite-paris-saclay.fr; 2Faculté des Sciences, Université Paris-Saclay, 91405 Orsay, France

**Keywords:** Ca^2+^ signalling, sigma-1 receptor, cancer

## Abstract

Ca^2+^ signaling plays a pivotal role in the control of cellular homeostasis and aberrant regulation of Ca^2+^ fluxes have a strong impact on cellular functioning. As a consequence of this ubiquitous role, Ca^2+^ signaling dysregulation is involved in the pathophysiology of multiple diseases including cancer. Indeed, multiple studies have highlighted the role of Ca^2+^ fluxes in all the steps of cancer progression. In particular, the transfer of Ca^2+^ at the ER-mitochondrial contact sites, also known as mitochondrial associated membranes (MAMs), has been shown to be crucial for cancer cell survival. One of the proteins enriched at this site is the sigma-1 receptor (S1R), a protein that has been described as a Ca^2+^-sensitive chaperone that exerts a protective function in cells in various ways, including the modulation of Ca^2+^ signaling. Interestingly, S1R is overexpressed in many types of cancer even though the exact mechanisms by which it promotes cell survival are not fully elucidated. This review summarizes the findings describing the roles of S1R in the control of Ca^2+^ signaling and its involvement in cancer progression.

## 1. Introduction

Calcium (Ca^2+^) plays a fundamental role as a second messenger, contributing to the regulation of a variety of physiological processes such as cell proliferation, metabolism, cell division and differentiation, gene transcription, cell motility, muscle contraction, secretion, programmed cell death, and neurotransmission [1].

The Ca^2+^ concentration inside the cytosol is tightly regulated and is maintained by a series of Ca^2+^ channels, pumps, and exchangers [2]. Moreover, cells tightly control their changes in cytosolic Ca^2+^ concentration by maintaining steep Ca^2+^ gradients with the extracellular medium, but also with cellular organelles. The endoplasmic reticulum (ER) is considered as the major intracellular Ca^2+^ store.

Different types of stimuli result in changes in Ca^2+^ levels inside the cell and subsequently in the ability of the cells in decoding them to give rise to several cellular outcomes. Ca^2+^ signals are modulated and decoded differently according to space, time, and amplitude [1,3].

Alterations in the regulation of Ca^2+^ signaling are involved in the pathogenesis of several diseases including diabetes, pathogen infections, neurodegenerative diseases, and also cancer [4]. Given the widespread role of Ca^2+^ signaling in healthy and cancer cells, Ca^2+^ fluxes have not much been considered as potential selective targets in the context of cancer therapy. Several recent studies have revealed a key role for Ca^2+^ in several steps involved in malignant transformation such as cell proliferation, cell migration, metastasis formation, modulation of the tumor microenvironment, and sensitivity to treatment [4,5]. Thus, even if calcium per se cannot constitute a therapeutic target, the proteins regulating calcium fluxes involved in these phenomena constitute good candidates for therapeutic treatment in cancer patients.

One important site for Ca^2+^ regulation is the interface between the ER and the mitochondria, the so-called mitochondrial associated membranes (MAMs). These structures participate in the modulation of Ca^2+^ dynamics, lipid synthesis, and transport, and other cellular events such as cell survival, mitochondrial fission, cell metabolism, and autophagy [6]. MAMs are enriched in Ca^2+^ channels and therefore tightly regulate this type of signaling between the ER and the mitochondria. Furthermore, it has been shown that they play a role in the modulation of cancer progression by targeting many proteins that have oncogenic or tumor-suppressive roles. How these proteins mediate cancer cell survival and tumor progression by altering ER mitochondrial Ca^2+^ signaling has been the focus of extensive investigation [7].

The sigma-1 receptor (S1R), an ER-resident membrane protein, is one of the many proteins that are enriched at MAMs. S1R has been proposed to have a chaperone activity and its action can be modulated by ligand binding. Moreover, S1R has been shown to play a cytoprotective role against cellular stress and to modulate Ca^2+^ signaling at the ER-mitochondria contact sites by regulating inositol-1,4,5-triphosphate (IP_3_ receptors-IP_3_R) [8]. Through its ability to bind and regulate different classes of proteins it acts as a pluripotent modulator to direct cell fate [9]. Interestingly, the S1R has also been shown to be involved in cancer cell physiology and to be overexpressed in a variety of tumors, suggesting that its protective role promotes cancer cell survival and tumor progression. In this review, we focus on the role of S1R in cancer via its involvement in the control of Ca^2+^ dynamics. Considering its high level of expression in many types of tumors, its impact on Ca^2+^ homeostasis, especially at the level of MAMs, could affect cell fate and disease pathogenesis. We thus suggest that the close relation between S1R, Ca^2+^ regulation at MAMs, and cancer could be a topic for a new field of research. Extensive research is needed to better characterize the role of S1R in this context.

## 2. Ca^2+^ Homeostasis Regulation in the ER

In order to be able to regulate several physiological functions, cells need to maintain a much higher Ca^2+^ concentration in the ER than in the cytosol. They have therefore developed a series of mechanisms that involve the action of Ca^2+^ buffering proteins, releasing and importing processes to maintain the high steady-state concentration inside this organelle [3,10,11]. The level of free Ca^2+^ inside the ER is in the µM range but the total Ca^2+^ concentration is estimated to be in the mM range [12]. This is the result of the high buffering capacity of the ER which is obtained by a series of Ca^2+^-binding chaperones such as calnexin, calreticulin, glucose-regulated protein/immunoglobulin heavy chain binding protein (GRP78/BiP), GRP94, and the protein disulfide isomerase PDI [13].

Calcium is thus actively accumulated in the ER by the ATP-dependent action of the sarco/endoplasmic reticulum Ca^2+^ ATPase (SERCA) proteins that transport cytoplasmic Ca^2+^ in the ER lumen against the Ca^2+^ concentration gradient. SERCAs are the only proteins that transport Ca^2+^ into the ER and are detected in all ERs. Thus, SERCAs along with other Ca^2+^ channels/pumps tightly regulate cellular Ca^2+^ homeostasis and are subjected to tight regulation [14]. The calcium stored in the ER can be released via IP_3_R, the ryanodine receptor (RyR), or by a calcium leak [15,16,17]. It is accepted that the IP_3_R is primarily responsible for calcium release from the ER in non-excitable cells. IP_3_Rs are ubiquitously expressed and are localized in the intracellular membranes, particularly in the ER. Cells can express three different isoforms of IP_3_R (IP_3_R1, IP_3_R2, and IP_3_R3) which form large tetrameric channels and are characterized by different affinities for IP_3_ [15]. IP_3_R activity is modulated by numerous factors such as the ER environment (pH, redox state, ATP, Ca^2+^, and Mg^2+^ concentrations), its phosphorylation status, and several regulatory proteins, including the S1R [18,19,20]. Ca^2+^ release from the ER through IP_3_R s most often occurs at MAMs, leading to mitochondrial Ca^2+^ uptake via the mitochondrial Ca^2+^ uniporter (MCU) [21].

In addition, the ER membrane is characterized by an inherent Ca^2+^ leakage. The mechanisms responsible for this constant leakage have not yet been fully elucidated but several proteins have been considered as candidates, such as members of the transmembrane BAX inhibitor motif-containing (TMBIM) family [22,23] or the translocon-ribosome complex [24,25,26].

A decrease in luminal Ca^2+^ concentration is sensed by the stromal interacting molecule (STIM) proteins that localize at the membrane of the ER. Upon store depletion, they oligomerize and redistribute into “puncta” at the ER-PM contact sites where they associate with and activate the ORAI channels to induce Ca^2+^ entry (reviewed in [27]).

## 3. The Mitochondria Associated Membranes

MAMs are very specialized structures that participate in the exchange of molecules such as lipids and Ca^2+^ between the two adjacent organelles. Proteomics studies have demonstrated that MAMs are highly enriched in proteins [28,29,30] that participate and regulate several cellular processes such as inflammation, lipids synthesis and trafficking, apoptosis, autophagy, ER stress, and Ca^2+^ handling [6,11,31,32,33]. This last function is highly important for cellular homeostasis because the ER and the mitochondria play a primary role in the modulation of Ca^2+^ homeostasis and because Ca^2+^ is vital for mitochondrial function [34]. Indeed, Ca^2+^ is required for promoting the activity of the enzymes of the tricarboxylic acid (TCA) cycle and subsequently for the correct production of ATP and maintenance of cellular bioenergetics. Cells lacking IP_3_R-dependent Ca^2+^ release activity induces a pro-survival autophagic response via AMPK activation [35]. In addition, mitochondrial Ca^2+^ uptake and subsequent overload are involved in the induction of programmed cell death via the action of the PTP, which induces membrane permeabilization and release of the matrix content [36].

Having such important consequences for cell homeostasis and cell fate, modulating Ca^2+^ exchanges in these microdomains is crucial. Among other proteins like ERp44 and Ero1α [37,38], S1R seems to play an important role in regulating ER-mitochondria contacts and mitochondrial dynamics [39].

Indeed, MAMs are highly dynamic structures that are able to adapt and modify their organization in order to overcome cellular stress [40,41]. This process is necessary to maintain homeostasis in physiological conditions, but it also plays a role during pathogenesis. One of the conditions during which ER-mitochondrial coupling is commonly altered is ER stress, a cellular state in which the protein folding capacity of the ER is decreased. In order to counteract such alteration, cells have developed an adaptive response that involves the activation of a network of signaling pathways commonly known as the unfolded protein response (UPR). This adaptive response, if the stress is prolonged in time, can lead to the induction of apoptosis [42]. It has been shown that during early phases of ER stress, the interactions between the ER and mitochondria are increased, affecting Ca^2+^ transfer and regulation of cellular bioenergetics [43,44]. Along this line, several chaperones and also UPR sensors have been identified in the MAMs (reviewed in [11]). It has been shown that the UPR sensor PKR-like ER kinase (PERK) is enriched in MAMs where it is required for correct Ca^2+^ transfer and maintenance of strong ER-mitochondria contact sites. Mechanistically, PERK acts as a structural tether that regulates inter-organellar communication and the cell response to ROS-induced ER stress-mediated cell death [45]. In addition, it has been shown that PERK activity and induction of UPR is modulated by another important MAMs component, mitofusin 2 (Mfn2). Mfn2 inhibits PERK activation through physical interaction impacting on mitochondrial morphology, ROS production, and mitochondrial respiration [46].

Inositol-requiring enzyme 1α (IRE1α), another UPR sensor, is enriched at the ER-mitochondria contact sites [47], where it acts as a scaffold to regulate the distribution of IP_3_Rs at MAMs, thereby impacting on mitochondrial Ca^2+^ uptake, ATP production, and metabolism. This impact on cellular bioenergetics is independent from its catalytic activity as a UPR transducer [48].

In recent years, dissection of MAMs dynamics has increased our knowledge on the correlation between MAMs dysfunction and human diseases. Changes in the normal communication between the ER and the mitochondria leads to metabolic defects and highly impacts on disease pathogenesis such as immune system activation [49], metabolic diseases [50], or several neurodegenerative disorders such as Alzheimer’s disease, Parkinson’s disease, and amyotrophic lateral sclerosis (ALS) [51].

MAMs dysfunction is also strongly correlated with the enhancement of cancer growth and metabolism. This is strongly mediated by the MAMs regulation of Ca^2+^ fluxes between the ER and mitochondria which can impact on ATP production, generally altered in tumors, and on the ability of cancer cells to undergo apoptosis. Indeed, many of the proteins tethering ER to mitochondria are oncogenes or tumor suppressors and have an impact on cancer cell survival by controlling Ca^2+^ transfer to mitochondria [52]. Overall, cancer cells exploit multiple strategies to control Ca^2+^ fluxes between the ER and mitochondria in order to promote cell survival and counteract apoptosis. This is obtained by a modulation of both ER Ca^2+^ release and mitochondrial Ca^2+^ uptake [7]. In this context, because of its impact on both Ca^2+^ fluxes in MAMs and cancer, it is important to underline the role of the sigma-1 receptor.

## 4. The Sigma-1 Receptor

The S1R is an integral membrane protein of the ER that is specifically enriched in the MAMs [20]. It has been discovered in the 1970s [53] but it was cloned for the first time in the 1990s when it was shown that this 223 amino acids-long protein shares no homology with any other mammalian protein [54]. It has always been considered as a receptor and several specific small molecules of synthetic and endogenous origins have been shown to be able to bind on this protein, triggering or inhibiting its biological responses [55].

Several studies have tried to characterize its structure, but the results are still controversial. It has been reported that S1R is able to form oligomers in the ER membrane and has a single transmembrane domain [56,57]. However, other studies support the hypothesis of the existence of two transmembrane domains [58,59], which has been confirmed by a 3D model using homology techniques [60,61]. The results concerning the localization of the C-terminus of the protein are also contradictory [57,62]. What is clear is that the oligomerization status of S1R is regulated by the binding to its ligands: binding to “agonists” such as (+)-pentazocine favors oligomer dissociation, while “antagonist” binding (e.g., haloperidol) promotes their stabilization [63,64].

Interestingly, the S1R is particularly enriched in the central nervous system (CNS) but its expression is relatively high also in other tissues such as liver, lung, and cardiac tissue. Much evidence has underlined the value of S1R as a modulator of cellular signaling and its key role in cytoprotection especially in a neurological context, promoting the use of its ligands as therapeutic agents for the treatment of neurodegenerative diseases [65,66].

The role of S1R in cytoprotection seems to be mediated by its action at the level of the MAMs, where it impacts on ER function and homeostasis. In this context, it has been reported that S1R directly binds to the ER-resident molecular chaperone GRP78/BiP but not to other ER chaperones [20]. The binding occurs between the C-terminal domain of S1R and the nucleotide-binding domain of BiP, but not on its substrate-binding domain [67], implicating that S1R is not a substate of this chaperone. This is confirmed by the fact that the S1R/BiP complex has a long stability and can therefore not be a mere interaction with S1R nascent proteins [20]. This protein association is affected by ligands able to bind S1R. Thus (+)-pentazocine favors dissociation of the complex, a process that is inhibited by haloperidol [20,64]. These results suggest that S1R oligomerization and subsequent inactivation is mediated by BiP binding.

Of note, the cytoprotective role of S1R is supported by data showing the ability of its ER-luminal domain to suppress the formation of protein aggregates, suggesting an activity similar to molecular chaperones [20]. Upon ER Ca^2+^ depletion, which is one of the causes of the induction of ER stress, S1R is able to bind to IP_3_R3, stabilizing it after dissociation from BiP and affecting Ca^2+^ uptake by the mitochondria [20]. In addition, S1R has a function during ER stress [68]. It has been observed that upon ER stress induction, S1R is upregulated via the PERK/eIF2α/ATF4 pathway [69] and that its overexpression suppresses the activation of UPR stressors PERK and ATF6 [20]. On the contrary, during ER stress S1R is able to promote and regulate IRE1α phosphorylation and activation and favor its stability at the MAMs. This action has an impact on the IRE1α/XBP1 pathway and on cell survival upon prolonged ER stress [47]. S1R is able to modulate cell responses to ER stress and to regulate UPR activation as it has been demonstrated by both in vitro and in vivo studies (reviewed in [68,70]).

Moreover, it has been observed in different cell types that, following stimulation with stressors and agonists, S1R is able to translocate to the plasma membrane (PM). Once at the PM, it interacts with a plethora of proteins including ion channels, receptors, and kinases (reviewed in [9]). S1R would also be able to translocate to the nuclear envelope to recruit chromatin-remodeling factors [71]. In addition, it has been observed that S1R is present in nuclear inclusions of neurons from patients with different proteinopathies, suggesting that S1R might shuttle from the cytoplasm to the nucleus where it promotes the degradation of nuclear inclusions by the ER-associated degradation machinery [72].

Overall, current research underlined the importance of the S1R interactome for the modulation of several cellular processes suggesting a role for S1R as a pluripotent modulator of cellular homeostasis. The dysfunction could represent a cause for the pathogenesis of multiple diseases, suggesting that targeting S1R may be a therapeutic opportunity for pathology treatment.

## 5. Sigma-1 Receptor in Cancer

S1R has been linked to the pathogenesis of cancer, although its role in this context has not been fully described yet. Indeed, S1R is overexpressed (as the sigma-2 receptor, the other subtype of sigma receptors) in rapidly proliferating cells, in cancer cell lines, and in tumor tissues [73,74]. Moreover, exogenous S1R ligands have cytotoxic and non-proliferative effects on tumoral cell lines [75,76,77].

It has been shown that lung, breast, and prostate metastatic cell lines have increased mRNA and protein levels of S1R, which correlate with the aggressiveness of the tumor [74]. Knockdown of S1R in breast cancer cell lines results in the reduction of cell proliferation and adhesiveness [74]. S1R is highly expressed in esophageal squamous cell carcinoma (ESCC) cell lines and it is enriched in samples derived from ESCC patients compared to normal tissue, which correlated with the severity of the tumor [78]. Moreover, pituitary tumors display an increased expression of S1R compared to normal epithelium [79]. A more recent study revealed an S1R overexpression in hepatocellular adenomas (HCA) compared to non-tumoral liver and demonstrated that this is mediated by estrogen receptor (ERα) transcriptional activity and correlates with loss of function of transcription repressor hepatocyte nuclear factor 1a (HNF1α). Indeed, S1R overexpression is higher among the HCA subtypes that present bi-allelic mutations of HNF1α, which is mostly found in women that are under treatment with estrogen [80].

On the contrary, immunohistochemical analysis of breast cancer samples revealed that the absence of S1R correlated with poorer disease-free survival, suggesting that S1R may play a role in inhibiting tumor growth [81]. The authors also revealed a positive correlation between S1R levels and the progesterone-receptor status. In addition, a recent study showed that S1R expression is decreased in hepatocellular carcinoma and inversely correlates with tumor grade. Overexpression of S1R in a hepatoblastoma cell line resulted in inhibition of cell proliferation, migration, and in the induction of apoptosis [82].

Even though it is still controversial, the enrichment of this receptor in many types of cancer cells opened to the possibility of the use of radiolabeled ligands for diagnostic purposes and for different types of imaging assays (reviewed in [83]). This characteristic of selective overexpression pinpoints S1R as a perfect target for drug delivery for specific anticancer treatment. Indeed, many studies have used S1R ligands for drug targeting by conjugating these ligands with nanoparticles containing cytotoxic drugs, antisense RNAs, or antitumor peptides (reviewed in [83]). This method showed promising results for targeted anticancer therapy but for now only in preclinical studies. Notably, since S1R plays a cytoprotective role, its ligands could be used as anticancer drugs in chemotherapy alone or in adjuvant therapy. Several studies have shown their ability in inducing growth inhibition and cell death induction in cancer cells in in vitro and in vivo experiments [84].

Even though the cytostatic and cytotoxic effects of S1R ligands have long been demonstrated, the mechanisms by which tumor growth inhibition and cell death are induced are not yet fully elucidated.

It has been demonstrated that S1R plays a role in the cap-dependent translation initiation in breast and prostate cancer cell lines. Treatment with S1R antagonists reduces the phosphorylation of regulatory proteins of translation in a reversible manner, suggesting the possibility of using S1R ligands as modulators of tumor cell protein synthesis [85]. S1R antagonist Rimcazole can promote cell death by inducing the hypoxia inducible factor-1*α* (HIF-1*α*) in colorectal and mammary carcinoma cells but not in non-tumoral cells. The induction of apoptosis is favored by the presence of p53 and the induction of a pro-apoptotic cell program [86]. In addition, it has been shown that S1R antagonists induce ER stress and UPR activation in adenocarcinoma cell lines. This results in the induction of autophagic flux and subsequently to apoptosis activation, indicating that UPR and autophagy induction mediate the cytoprotective role of S1R in cancer cells [87]. Another work has underlined the role of S1R in contributing to the interleukin-24 (IL-24)-mediated induction of apoptosis via IL-24-S1R interaction, resulting in diminished ER stress induction, ROS production, and Ca^2+^ mobilization [88].

In hepatocellular adenomas S1R promotes not only proliferation but also lipid accumulation, suggesting that it is involved in the induction of steatosis, a peculiar characteristic of this type of tumor [80]. Indeed, MAMs are an important site for lipid metabolism and the S1R receptor has been shown to play a role in this context by mediating the induction of lipogenesis [89,90].

Extensive research has focused on the role of S1R in modulating the activity of ion channels such as voltage-dependent channels, volume-regulated chloride channels, acid-sensing ion channels, and N-methyl-D-aspartic acid receptor (NMDA). Ion channels are crucial not only in the regulation of cell signaling but also for ion and water homeostasis. Their activity and electrical plasticity are remodeled in cancer cells and play a role in tumor progression [91].

A voltage-dependent K^+^ channel human ether-à-go-go-related gene (hERG) is considered as a biomarker for many solid tumors and acute or chronic leukemias [92,93,94,95]. S1R mediates the recruitment of hERG at the plasma membrane and their association with β1-integrin in leukemic and colorectal cancer cells [96]. This in turn suppresses the activation of the Akt pathway and the subsequent cell migration by acting on cytoskeleton remodeling, with an effect on cell invasiveness in in vitro and in vivo experiments [96]. In addition, S1R associates with the Nav1.5 voltage-gated Na^+^ channel via their transmembrane regions [97]. This interaction is crucial for the regulation of ion currents that have been shown to promote the invasiveness of breast cancer cells [98]. Activation of S1R in lung cancer and leukemia cells results in the inhibition of chloride currents and cell proliferation that provokes a cell cycle arrest in the G_1_ phase. This effect is mediated by a functional coupling of S1R with volume-regulated chloride channels (VRAC), implying that S1R participates in the regulation of cell volume, a fundamental event that modulates the cell cycle and apoptosis [99].

Interestingly, S1R has been shown to be a modulator of Ca^2+^ homeostasis. S1R ligands cause dysregulation of cytosolic Ca^2+^ transients thereby affecting cardiomyocyte contractility [100] but also inducing an alteration of Ca^2+^ responses to N-methyl-D-aspartate (NMDA) in rat cortical neurons [101,102]. Agonist treatment is able to increase the Ca^2+^ influx in rat hippocampal neurons via PKC activation [103]. In addition, it has been shown that, in mouse neuroblastoma cells, S1R agonists induce an ER Ca^2+^ release by promoting the dissociation of adaptor protein ankyrin B 220 (ANK 220) from IP_3_R3, releasing it from inhibition [104]. This seems to be mediated by the C-terminal part of S1R [105]. This in turn affects Ca^2+^ fluxes at the interface between the ER and the mitochondria, where S1R reduces the degradation of IP_3_R3, whose activation in turn promotes S1R-BiP dissociation. S1R knockdown reduces the uptake of Ca^2+^ by the mitochondria at the level of the MAMs, also affecting functional recovery of IP_3_R after repetitive stimuli [20].

Furthermore, in hepatocytes, S1R agonists are able to inhibit Ca^2+^ oscillations triggered by induction of IP_3_ synthesis. Notably, agonist treatment reduces IP_3_ production via activation of conventional PKC, counteracting the signaling pathway responsible for the transduction of the stimulus into IP_3_ production [106].

Overexpression of S1R has been shown to inhibit SOCE by decreasing the coupling between the store depletion and Ca^2+^ influx. Indeed, binding to STIM1 S1R reduces the rate of translocation of STIM1 to the plasma membrane and the subsequent binding to ORAI1. This phenomenon is modulated by S1R ligands [107].

Since increasing evidence supports the crucial role of Ca^2+^ in the modulation of the pathogenesis of cancer [4] and because S1R is enriched in many types of tumors where it often provides a cytoprotective function, it can be expected to perform its pro-tumorigenic function by the regulation of Ca^2+^ homeostasis inside the cancer cells.

Along this line, Spruce and colleagues have demonstrated that the pro-survival effect of S1R in tumoral cells is abolished by treatment with S1R antagonists, activating a pro-apoptotic response. This seems to be induced by an increase in cytosolic Ca^2+^ which activates phospholipase C (PLC) [76]. A more recent work explored the link between S1R and plasma membrane Ca^2+^ signaling in cancer cells. The authors have shown that, in breast and colon cancer cells, S1R associates with calcium-activated K^+^ channel SK3 (KCNN3), mediating their coupling to ORAI1 Ca^2+^ channels in caveolae lipid nanodomains. This results in reduced Ca^2+^ influx and subsequent migration inhibition [108]. A lot of work has yet to be done to improve our knowledge of the role of S1R in the modulation of Ca^2+^ dynamics in cancer cells but this would lead to an important contribution in the characterization of the molecular mechanisms by which S1R affects cancer cell homeostasis.

## 6. Conclusions

Due to its ability to bind several types of proteins and exert different functions inside the cell, S1R begins to be seen as a pluripotent modulator of cell signaling (Figure 1).

Due to this characteristic, S1R is involved in a variety of pathologies and conditions such as neurodegenerative diseases and drug addiction. Notably, S1R is overexpressed in many types of cancer cells and tumoral tissues, even though this observation remains controversial. The role of S1R in pathophysiology has been intensively studied in the cancer context. Nonetheless, its mechanism of action has not yet been fully characterized. Many studies have explored its effect on the modulation of ion channel activity in the tumoral context and on the electrical plasticity of cancer cells. Indeed, aberrant ion channel activity is one key property of tumoral cells and among these Ca^2+^ flux dysregulation has an impact on several steps in cancer progression. The studies that we have summarized in this review indicate accumulating evidence of the role of S1R in the modulation of Ca^2+^ homeostasis, although this impact in the context of cancer cells has not been intensively explored. Further studies are required to better highlight the function of S1R in this context. Due to its ability to bind different ligands that modulate its action inside the cell, S1R could be also considered as a promising pharmacological target for therapeutic intervention in the context of cancer treatment. Therapeutic modulation of S1R in cancer therapy could represent a novel strategy for regulating Ca^2+^ homeostasis and thereby impact on cancer cell survival and cancer progression.

## Figures and Tables

**Figure 1 genes-12-00139-f001:**
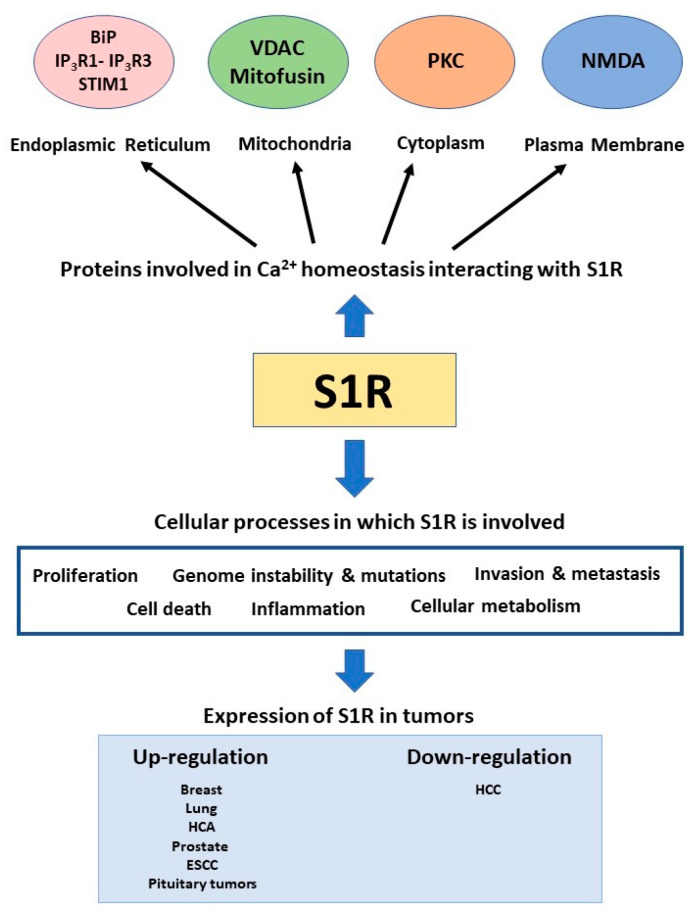
Summary of the involvement of S1R in calcium signaling and cancer.

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
