# Peer review of "Role of Sigma-1 Receptor in Calcium Modulation: Possible Involvement in Cancer"

_genes, 2021, doi:10.3390/genes12020139_

Round 1

Reviewer 1 Report

Pontisso and Combettes presented a comprehensive review on the topic of Sigma-1 receptor (S1R) in calcium modulation, with a focus on its role in Mitochondrial Associated Membranes (MAMs). The review also has a clinical perspective with in depth discussion of S1R involvement in cancer.

The review was very well organized and the its content is of high relevant to several fields including but not limited to: calcium regulation, ER and mitochondrial physiology and oncology. 

In my opinion, the review is well done and no major revision is needed. However, the language might need further polish from place to place. Below is a non-exhausted list of corrections that need to be fixed:

L9: Ca2+ signalling plays a pivotal role in the control of cellular homeostasis, and therefore aberrant regulation of Ca2+ fluxes have strong impact on cellular functioning.

L47: Important sites for Ca2+ regulation are the interface between the ER and the mitochondria, the so-called Mitochondrial Associated Membranes (MAMs).

Suggested changes: One important site for Ca2+ regulation is the ....

L49: These structures participate in (not to) the modulation of Ca2+ dynamics...

L89: It is accepted that the IP3R is primarily responsible for calcium release from the ER in non-excitable (cells).

L185: but its expression is enhanced (I suggest wording differently, eg: is relatively high) also in other tissues such as liver, lung and cardiac tissue.

Author Response

Dear Sir,

We are happy that the reviewer found our review interesting and we thank him for these constructive comments.

The corrections requested by the reviewer have been made as well as other language corrections seen during the proofreading. All these corrections appear in the new version of our article.

L9: Ca2+ signalling plays a pivotal role in the control of cellular homeostasis, and therefore aberrant regulation of Ca2+ fluxes have strong impact on cellular functioning.

This modification has been made

L47: Important sites for Ca2+ regulation are the interface between the ER and the mitochondria, the so-called Mitochondrial Associated Membranes (MAMs).

Suggested changes: One important site for Ca2+ regulation is the ....

This change has been made

L49: These structures participate in (not to) the modulation of Ca2+ dynamics...

This error has been corrected

L89: It is accepted that the IP3R is primarily responsible for calcium release from the ER in non-excitable (cells).

This error has been corrected

L185: but its expression is enhanced (I suggest wording differently, eg: is relatively high) also in other tissues such as liver, lung and cardiac tissue.

This change was made as suggested by the reviewer

Reviewer 2 Report

This is a comprehensive and very interesting review on a topic of increasing importance in recent years.

1) p.2, line 89, "...release from the ER in non-excitable..." "cells" is missing.

2) p.7, lines 336-339. "A lot of work...of the molecular mechanisms..." should be deleted since it is repeated once more and the sentence finished.

3) The list of references has to be studied carefully and corrected where appropriate

Author Response

We are happy that the reviewer found our review interesting and we thank him for these constructive comments. All the corrections requested by the reviewer have been made in the new version of our manuscrit.

1) p.2, line 89, "...release from the ER in non-excitable..." "cells" is missing.

This error has been corrected

2) p.7, lines 336-339. "A lot of work...of the molecular mechanisms..." should be deleted since it is repeated once more and the sentence finished.

As suggested by the reviewer, this repetition has been deleted.

3) The list of references has to be studied carefully and corrected where appropriate

We have carefully checked the list of references and changed the format of this list in accordance with the format requested by the journal.